# Unveiling the Significance of LysE in Survival and Virulence of *Mycobacterium tuberculosis*: A Review Reveals It as a Potential Drug Target, Diagnostic Marker, and a Vaccine Candidate

**DOI:** 10.3390/vaccines12070779

**Published:** 2024-07-16

**Authors:** Shilpa Upadhyay, Archana Dhok, Supriya Kashikar, Zahiruddin Syed Quazi, Vinod B. Agarkar

**Affiliations:** 1Global Consortium of Public Health Research, Jawaharlal Nehru Medical College, Datta Meghe Institute of Higher Education and Research, Sawangi, Wardha 442107, Maharashtra, India; 2i-Health Consortium, Jawaharlal Nehru Medical College, Datta Meghe Institute of Higher Education and Research, Sawangi, Wardha 442107, Maharashtra, India; drarchanadhok@gmail.com; 3GeNext Genomics Pvt. Ltd., Nagpur 440010, Maharashtra, India; supriyakashikar@genextgenomics.com (S.K.); vinodagarkar@genextgenomics.com (V.B.A.); 4Global Evidence Synthesis Initiative, Jawaharlal Nehru Medical College, Datta Meghe Institute of Higher Education and Research, Sawangi, Wardha 442107, Maharashtra, India; zahirquazi@gmail.com

**Keywords:** lysine exporter, *Mycobacterium tuberculosis*, LysE, transmembrane protein, lysine, arginine

## Abstract

Tuberculosis (TB) remains a global health threat, necessitating innovative strategies for control and prevention. This comprehensive review explores the *Mycobacterium tuberculosis* Lysine Exporter (LysE) gene, unveiling its multifaceted roles and potential uses in controlling and preventing tuberculosis (TB). As a pivotal player in eliminating excess L-lysine and L-arginine, LysE contributes to the survival and virulence of *M. tuberculosis*. This review synthesizes findings from different electronic databases and includes 13 studies focused on the LysE of *M. tuberculosis*. The research unveils that LysE can be a potential drug target, a diagnostic marker for TB, and a promising candidate for vaccine development. The absence of LysE in the widely used BCG vaccine underscores its uniqueness and positions it as a novel area for TB prevention. In conclusion, this review underscores the significance of LysE in TB pathogenesis and its potential as a drug target, diagnostic marker, and vaccine candidate. The multifaceted nature of LysE positions it at the forefront of innovative approaches to combat TB, calling for sustained research efforts to harness its full potential in the global fight against this infectious disease.

## 1. Introduction

Tuberculosis (TB) stands as a formidable and pervasive health threat, ranking as the 13th leading cause of death worldwide [1]. Before the emergence of the COVID-19 pandemic, tuberculosis stood as the leading cause of mortality attributable to a single infectious agent, surpassing even the impact of HIV/AIDS [2]. In 2022, it caused 1.3 million deaths globally, with the WHO South-East Asian Region seeing a surge in new cases (46%), followed by the African Region (23%) and the Western Pacific (18%) [3]. Developing countries, including high TB-burdened nations like India, face challenges such as inadequate diagnosis, treatment non-adherence, increasing drug-resistant cases, and low awareness, leading to a rapid rise in TB [3]. Tackling these issues is vital to curb TB’s impact and strengthen global health resilience.

Drug-resistant TB remains a persistent challenge to public health. The WHO employs five categories to classify drug-resistant cases: isoniazid-resistant (IR) TB, rifampicin-resistant (RR)-TB, multidrug-resistant TB (MDR-TB), extensively drug-resistant TB (XDR-TB), and pre-XDR-TB. MDR-TB, defined by resistance to both rifampicin and isoniazid, and XDR-TB, resistant to rifampicin, plus certain second-line drugs, are especially worrisome. Pre-XDR-TB denotes resistance to rifampicin and any fluoroquinolone [4]. In 2022, an estimated 410,000 people developed MDR/RR-TB globally, highlighting the urgent need for action to address these challenges [4]. MDR-TB and XDR-TB represent significant obstacles to achieving the End TB Strategy and the UN Sustainable Development Goals, necessitating immediate attention and intervention [4,5]. In countries with a high burden of TB, such as India, Pakistan and China, there has been a concerning surge in the incidence of MDR-TB and XDR-TB, indicating a lack of effective control measures [4]. In these regions, the diagnosis of MDR-TB, XDR-TB, and other forms of drug-resistant TB, as well as TB associated with HIV, can pose challenges due to their complexity and the associated high costs [3].

For the prevention of TB, the only available Bacille Calmette-Guérin (BCG) vaccine has been a cornerstone of immunization for over eight decades, reaching over 80% of newborns and infants in countries with integrated childhood vaccination programs [6]. It demonstrates efficacy in protecting children from meningitis and disseminated tuberculosis, yet it does not prevent initial infection or the reactivation of latent pulmonary infection, limiting its impact on *Mycobacterium tuberculosis* transmission [7]. Thus, while vital for childhood health, its role in curbing TB prevention and transmission dynamics remains somewhat constrained.

Recognizing this limitation, there is a critical need to explore new vaccine candidates that are absent in BCG and can induce host immunity. This may help in the prevention and transmission of this deadly disease. *Mycobacterium tuberculosis* (*M.tb*) possesses different proteins or antigens that are absent in BCG and possess the potential to induce specific humoral, cell-mediated, and innate responses in the host [8].

In this section below, we present a brief overview of *M.tb* along with its pathogenicity and virulence. In addition, this review focuses on a less explored amino acid transporter protein of *M.tb* that is absent in BCG and has the potential to be a novel vaccine candidate.

### 1.1. Overview of Mycobacterium tuberculosis

*Mycobacterium tuberculosis (M.tb)* is an aerobic, slow-growing, rod-shaped, non-motile, chemoorganotrophic, non-spore-forming bacillus, initially discovered by Robert Koch in 1882 and remains a significant focus of research and intervention efforts [9,10]. It serves as the prototype within the *M. tuberculosis* complex (MTBC), a group of mycobacterial pathogens responsible for tuberculosis (TB) in various mammalian species. MTBC is composed of several highly genetically related species (>99% nucleotide identity). Alongside *M. tuberculosis*, the MTBC encompasses other members such as the human-specific *M. africanum*, as well as animal-adapted strains including *M. bovis*, *M. caprae*, *M. pinnipedii*, *M. mungii*, *M. orygis*, and *M. microti* [10,11]. Of these members, *M. tuberculosis* is the leading infectious, highly adaptive intracellular pathogen causing tuberculosis in humans that destroys phagosomal cells to maintain and evade the immune system [9].

The cell wall of *M. tuberculosis* is a unique characteristic of the mycobacteria [12]. The structure resembles Gram-negative bacteria, featuring a second “outer membrane” that contains mycolic acids, which are lengthy, branched fatty acids [10]. Notably, mycolic acids serve as integral components that confer remarkable impermeability to the cell wall. This thick unique structural feature plays a pivotal role in the organism’s ability to withstand host defenses and environmental pressures, thus amplifying its pathogenic potential [13]. The outer layer surrounding the cytoplasmic membrane forms a significant complex, where the core consists of peptidoglycan connected to arabinogalactan, which is then linked to the mycolic acids (referred to as the mAGP complex) [10]. The intricate, thick, waxy cell wall is crucial for virulence and acts as a barrier, making it impermeable to antibiotics and several frontline anti-tuberculosis drugs (including isoniazid, ethambutol, and ethionamide) target its synthesis [10,11]. In the context of mycobacterial pathogenicity, it is essential to explore the distinctive lipid composition of the cell wall in *M. tuberculosis*. The structure of *M. tuberculosis* cell wall is depicted in Figure 1.

### 1.2. TB Pathogenesis

The pathogenesis induced by MTBC involves intricate interactions among the host immune system and the bacterial agents [9]. *M.tb* is a highly pathogenic human pathogen with a significant historical impact [10]. Transmission of *M.tb* occurs through the inhalation of aerosolized droplets from individuals with active pulmonary TB, whereby the bacilli are recognized by host macrophages via pathogen-associated molecular patterns (PAMPs) and pattern recognition receptors (PRRs) in the lungs [14]. Upon inhalation, tubercle bacilli invade the lungs, evade initial immune responses, replicate within alveolar macrophages, and disseminate to other tissues and organs via the bloodstream and lymphatics if innate defences are insufficient [9,11]. A cell-mediated immune response typically controls bacterial replication, resulting in asymptomatic latent TB in most cases, characterized by a dynamic equilibrium between bacilli and host immunity [11]. However, any compromise to cell-mediated immunity may trigger active bacterial replication, tissue damage, and the manifestation of active TB disease [11]. The steps involved in pathogenesis are depicted in Figure 2.

### 1.3. TB Virulence

*Mycobacterium tuberculosis* (*M.tb*), as an obligate intracellular pathogen, invests a significant portion of its proteome into proteins that enhance its survival [16]. Notably, *M.tb* diverges from conventional bacterial pathogens by lacking classical virulence factors like toxins, a trait shared with non-pathogenic mycobacteria [17]. Despite its stealthy nature, *M.tb* can persist within the host for extended periods, especially in latent tuberculosis scenarios, where it remains quiescent unless host immunity is compromised, such as in HIV-associated TB [18]. The evolutionary arms race between *M.tb* and the host’s immune system has led to the development of various virulence factors, which facilitate intracellular survival and support its lifestyle within granulomas during latency [17,18]. These factors encompass both proteinaceous and non-protein elements, such as lipids, glycans, glycolipids, and nucleic acids [16]. Foremost among *M.tb’s* virulence mechanisms are its protein secretion systems, with five type VII secretion systems identified, including the well-studied ESX1 [11]. Notably, ESX1 is absent in the attenuated *M. bovis* Bacille Calmette-Guérin (BCG) vaccine strain, suggesting its crucial role in virulence [11]. Additionally, the absence of regions RD1, RD2, and RD14 in *M. bovis* BCG appears to have occurred during or after the attenuation process, indicating their significance in virulence [17]. Furthermore, *M.tb’s* virulence repertoire includes transporter proteins, encompassing both importers and exporters, which contribute to its pathogenicity [17]. Understanding the intricacies of these virulence mechanisms is essential for developing effective strategies to combat tuberculosis.

### 1.4. LysE, an Amino Acid Transporter Protein

The survival and virulence of *M.tb* are likely influenced by various ion channels and transporter proteins that the bacterium possesses [19]. These transporter proteins play a crucial role in the physiological processes of *M.tb*, contributing to its ability to survive and cause disease [19]. In mycobacteria, L-Lysine exporters (LysE), a small transmembrane protein with a molecular weight ranging from 20 to 25 kDa, play a pivotal role in eliminating excess metabolically produced basic amino acids such as L-lysine and L-arginine and their toxic analogues from the cytosol [20]. The accumulation of these toxic analogues can lead to bacterial growth retardation, which reveals that LysE is an essential component for bacterial growth. The absence of these exporters results in elevated intracellular levels of L-lysine, impeding the growth of the bacteria [21].

LysE, an exporter of L-Lysine, has emerged as a promising candidate in various studies, highlighting its potential applications as a drug target, vaccine candidate, and prospective diagnostic marker [21,22,23,24]. LysE, also known as Rv1986, a member of RD2 antigens that are absent in the widely used BCG vaccine defines its uniqueness, and hence, LysE can be a suitable new vaccine candidate [8,22,23,24]. Therefore, comprehensive knowledge about this exporter is required for its consideration as a potential drug target for inhibition, diagnostic marker and probable vaccine candidate. Much research has been conducted on Lysine and arginine exporters of *Escherichia coli* (ArgO) and *Corynebacterium glutamicum* (LysE); however, the LysE of *Mycobacterium tuberculosis*, which is responsible for bacterial growth and able to stimulate innate and adaptive immunity, is still a novel area for research.

Therefore, this review article aims to highlight the challenges faced in TB control efforts and underscores the necessity for novel strategies. By investigating the function and significance of a protein called LysE in *M.tb*, this article suggests its potential as a target for new drugs, a promising candidate for vaccines, and a marker for TB diagnosis.

## 2. Materials and Methods

### 2.1. Inclusion and Exclusion Criteria

We searched various databases, including PubMed, Scopus, Web of Science, CENTRAL, and Google Scholar, using specific keywords related to the Lysine Exporter gene of Mycobacterium tuberculosis. Inclusion criteria involved selecting articles from full journal publications. We excluded articles containing information about the Lysine-Exporter gene from other bacterial species like *C. glutamicum*, *E. coli*, *H. pylori*, etc. However, articles that discussed LysE of *Mycobacterium tuberculosis* along with LysE of other bacterial species were included. The data extraction, however, focused solely on the LysE of *Mycobacterium tuberculosis*. We included the studies of the last 25 years. No language restrictions were applied. We performed the search between September and October 2023 and presented search strategies for PubMed, Scopus, CENTRAL and Web of Science here.

### 2.2. Search Strategy

#### 2.2.1. Pubmed

(((((((“lysine export*”[Title/Abstract]) OR (“lysine permea*”[Title/Abstract])) OR (LysE[Title/Abstract])) OR (“amino acid transport*”[Title/Abstract])) OR (“lysine efflux”[Title/Abstract])) OR (“lysine transport*”[Title/Abstract])) OR (Rv1986[Title/Abstract])) OR (lysine[MeSH Terms])

(((tubercul*[Title/Abstract]) OR (TB[Title/Abstract])) OR (mycobacteri*[Title/Abstract])) OR (“Mycobacterium tuberculosis”[MeSH Terms])

(#1) AND (#2)

#### 2.2.2. Scopus

TITLE-ABS-KEY(“lysine export*”) OR TITLE-ABS-KEY(“lysine permea*”) OR TITLE-ABS-KEY(“lysine efflux”) OR TITLE-ABS-KEY(“lysine transport*”) OR TITLE-ABS-KEY (LysE) OR TITLE-ABS-KEY(Rv1986) OR TITLE-ABS-KEY(“amino acid transport*”) AND TITLE-ABS-KEY(tubercul*) OR TITLE-ABS-KEY(mycobacteri*) OR TITLE-ABS-KEY(“Mycobacterium tuberculosis”) OR TITLE-ABS-KEY(TB).

#### 2.2.3. Web of Science

((((((KP=(“lysine export*”)) OR KP=(“lysine permea*”)) OR KP=(“lysine efflux”)) OR KP=(“lysine transport*”)) OR KP=(LysE)) OR KP=(Rv1986)) OR KP=(“amino acid transport*”)

((KP=(tubercul*)) OR KP=(mycobacteri*)) OR KP=(“Mycobacterium tuberculosis”)

#### 2.2.4. CENTRAL

Search Name: LysE_Mycobacterium tuberculosis

Comment: IDSearchHits#1“Lysine export*”0#2“lysine permea*”0#3“lysine transport*”1#4“lysine efflux”1#5LysE112#6“amino acid transporter”44#7MeSH descriptor: [Lysine] explode all trees and with qualifier(s): [biosynthesis − BI]1#8#1 OR #2 OR #3 OR #4 OR #5 OR #6 OR #7159#9tubercul*9065#10mycobacteri*2495#11TB7621#12“Mycobacterium tuberculosis”1073#13MeSH descriptor: [Mycobacterium tuberculosis] explode all trees415#14#9 OR #10 OR #11 OR #12 OR #1314,759#15#8 AND #140

## 3. Results

We initially identified 531 potential articles across various electronic databases, with the majority from PubMed (*n* = 258), Scopus (*n* = 271), and Web of Science (*n* = 2), along with two articles from other sources. Following the removal of duplicates, a total of 438 articles were examined. Through initial screening based on title and abstract, 414 articles were excluded. Subsequently, the full texts of 24 articles were independently assessed, leading to the exclusion of 11 articles that lacked specific details on mycobacterial LysE or focused on details of lysine exporters of other bacterial species. Ultimately, 13 articles were identified for potential inclusion. We present a PRISMA flow diagram detailing the selection of articles in Figure 3.

### Details of Included Studies

A total of 13 studies were included that describe the role of LysE in the bacterial system. Out of 13, the majority of the research on LysE was performed in China [8,24,25,26] and the United States of America [21,27,28,29], followed by Germany [30], Mexico [31], South Africa [23], United Kingdom [22] and in India [32]. We included the research of the last 25 years.

The included studies define the multifaceted role of LysE as a probable candidate for vaccine development, drug target and diagnostic marker. Five [8,21,22,23,25] of 13 studies described LysE’s role as a probable vaccine candidate, another five [26,27,28,29,32] explained their role as a promising drug target, and the remaining three studies [24,30,31] defined LysE as a diagnostic marker. Among 13 studies, three studies [8,22,23] defined LysE as both a new vaccine candidate and a diagnostic marker.

We share Table 1, summarizing the research conducted on LysE so far. It highlights LysE’s potential as a vaccine candidate, drug target, and diagnostic marker. This table calls for more research on LysE and provides a quick overview of the work performed in this field.

## 4. Discussion

The 2023 s United Nations high-level meeting on TB established ambitious global targets, notably prioritizing the development of new TB vaccines within a five-year timeframe, with a strong emphasis on safety and efficacy. This initiative is complemented by efforts to foster an enabling environment for TB research, fostering innovation and collaboration among Member States and partners. The urgency for intensified research and innovation is underscored by the End TB Strategy, which highlights the crucial role of vaccines in mitigating infection risk and disease progression [4]. As of August 2023, 16 vaccine candidates and 28 drugs are in various phases of clinical trials, aiming to prevent TB infection and disease while improving treatment outcomes [4]. Recognizing the pivotal role of TB vaccines in achieving rapid reductions in TB incidence and mortality, the WHO has initiated high-level actions, including establishing an “accelerator council” and a flagship initiative to accelerate progress towards ending TB [4].

These initiatives underscore a concerted effort to address the global TB burden through innovation and collaboration. However, alongside enhanced diagnostics and chemotherapy, the urgent need for a more efficacious vaccine is paramount to mitigate the morbidity and mortality associated with TB, particularly among vulnerable populations at heightened risk of active TB onset [16]. *M.tb’s* adeptness in evolving immune evasion mechanisms necessitates advancements in vaccines, diagnostics, and treatments to effectively combat this resilient pathogen [14]. Exploration into several amino acid biosynthesis pathways crucial for the survival and pathogenesis of *Mycobacterium tuberculosis* has shed light on their pivotal role in the replication machinery and cell wall synthesis of the bacterium. Notably, many of these pathways or their significant enzyme homologs are absent in humans and animals [33]. Due to their absence in humans and animals, they are considered a potential target for drug and vaccine development.

Among these pathways, the aspartate pathway synthesises essential amino acids such as lysine, isoleucine, threonine, and methionine, which is significant since it involves many essential enzymes for bacterial virulence and survival [28]. Lysine, an essential amino acid, is produced in mycobacteria via the aspartate pathway using Meso-diaminopimelate (DAP) as a direct precursor [34]. Meso-diaminopimelate (DAP) is a primary constituent of bacterial cell wall peptidoglycan) [34]. Excess lysine inside the bacterial cell leads to toxic effects on the cell resulting in growth retardation. Transmembrane proteins like L-lysine exporters (LysE) are vital in bacteria to remove excess L-lysine and L-arginine produced during metabolic processes from the cytosol. If these exporters are absent, cellular levels of L-lysine increase, impeding bacterial growth [21]. These lysine exporters are present on the inner membrane of bacterial cells, facilitating the efficient removal of surplus lysine from the cytosol.

L-lysine exporter (LysE) manifests as a small transmembrane protein with a molecular weight ranging from 20 to 25 kDa [20]. Functioning as a dimer, this protein exhibits a structural arrangement characterized by five to six transmembrane hydrophobic spanning helices [29]. The specific configuration of LysE underscores its role as a functional unit with distinct characteristics crucial for its cellular activities [35,36]. LysE was discovered primarily in *C. glutamicum* and later in *M. tuberculosis*, *B. subtilis*, *E. coli*, and *H. pylori* [26]. It was the first amino acid exporter to be molecularly recognized [35]. The LysE gene has sequence similarities to *E. coli*’s arginine exporter and *C. glutamicum’s* lysine exporter, ArgO, and LysE, respectively. It has been reported that LysG stimulates the transcription of lysE in the presence of lysine or histidine [30]. Ionic homeostasis, transmembrane electron transport, protection from high cytoplasmic heavy metal/metabolite concentrations, and cell envelope assembly are all functions of the members of the LysE Superfamily [26]. LysE is observed to facilitate the unidirectional removal of L-lysine, along with other vital amino acids like L-arginine, through efflux. It is recognized as the exclusive pathway for the excretion of L-lysine. The proton motive force (H+ antiport) is thought to be the energy source [37].

LysE becomes a crucial target for bacterial inhibition due to its essential role in preventing the accumulation of toxic analogues or toxic levels of L-lysine. A deficiency in LysE results in the build-up of these harmful substances, inhibiting bacterial growth [21]. As such, targeting LysE presents a promising strategy to impede bacterial proliferation and ensure the effective control of bacterial populations. A study reported [32] that L-canavanine (CAN) and L-thialysine (Thl) are the toxic analogues of lysine and arginine, respectively. Incorporation of these toxic analogues during protein synthesis leads to aberrant protein synthesis and ultimately leads to toxicity or growth inhibition [32]. LysE, identified as the Rv1986 gene in *Mycobacterium tuberculosis*, stimulates the proliferation of both innate and adaptive immune cells, promoting the release of cytokines. According to the cytokine secretion and T cell population data, it can potentially boost protective immunity via T cells, plasma cells, and inflammatory cytokines [8,31]. LysE belongs to the region of deletion 2 (RD2) antigens responsible for bacterial virulence and growth, which was lost between 1927 to 1931 when vaccinologists reported the further attenuation of the BCG vaccine [26,38].

L-lysine exporter (LysE) emerges as a multifaceted candidate in several studies, being recognized not only as a potential vaccine candidate [8,21,22,23,25] but also as a promising drug target [26,27,28,29,32] and a prospective diagnostic marker [24,30,31]. Its absence in the widely used BCG vaccine defines its uniqueness, and hence, LysE can be a suitable new vaccine candidate [8,22,23]. These findings highlight the versatile applications of LysE in the realms of vaccine development, therapeutic intervention, and diagnostic advancements, showcasing its significance across various biomedical contexts. To prevent the further development of multidrug-resistant strain, inhibition or blocking such efflux pumps may be considered an effective therapeutic strategy.

### Strengths and Limitations

This review is the first comprehensive study of LysE in *Mycobacterium tuberculosis* in the last 25 years. It gathers all existing research, setting a strong base for future studies. Despite a small pool of studies, it covers all relevant findings, giving a deep insight into LysE. It also pinpoints areas needing more exploration, guiding future research to fully understand LysE’s role in tuberculosis. Although faced with challenges like limited access to full-text articles, the review effectively uses available resources, like abstracts, to provide a thorough overview of LysE research.

The lack of research on LysE in *M.tb* poses a challenge to thoroughly analyzing certain aspects, potentially resulting in gaps in comprehension and interpretation. Relying solely on abstracts when full-text access is unavailable may limit the completeness and precision of data extraction, as abstracts might not capture the intricacies and discoveries of the original studies. Furthermore, including only accessible studies could unintentionally introduce bias by overlooking pertinent research published in journals or repositories with limited access.

## 5. Conclusions

This review underscores the ongoing challenges posed by tuberculosis (TB) and the urgent need for innovative approaches to combat this persistent global health threat. Despite advancements in diagnosis and treatment, TB remains a leading cause of mortality worldwide, with multidrug-resistant strains posing particularly concerning challenges. The exploration of *Mycobacterium tuberculosis* (*M.tb*) protein, LysE presents promising avenues for the development of novel therapeutics. According to the literature, LysE’s multifaceted role as an amino acid transporter protein highlights its potential as a crucial drug target, a prospective diagnostic marker, and a promising candidate for vaccine development. However, further research is warranted to fully elucidate LysE’s functional significance in *M.tb* pathogenesis and to translate these findings into tangible improvements in TB control strategies. Efforts to address the gaps in our understanding of LysE and its implications for TB management are essential to advance towards the ambitious goal of eradicating TB by 2030 outlined in the End TB Strategy.

## Figures and Tables

**Figure 1 vaccines-12-00779-f001:**
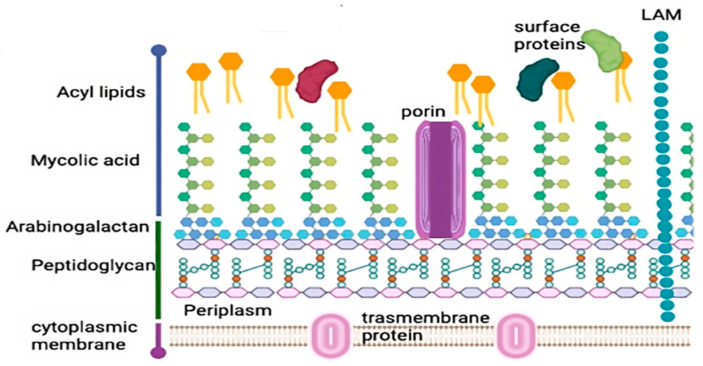
*M. tuberculosis* cell wall. The cell wall comprises an outer and inner membrane, delineating a periplasmic space. This space hosts a thin layer of peptidoglycan intricately linked to arabinogalactan and lipoarabinomannan (LAM), subsequently anchored to mycolic acids. The image has been adopted from a study conducted by Jacobo-Delgado 2023 [12].

**Figure 2 vaccines-12-00779-f002:**
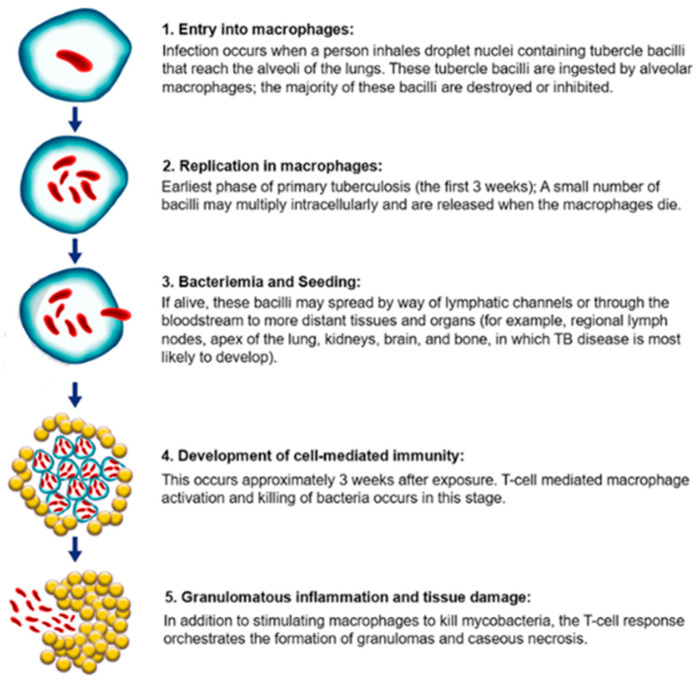
*M. tuberculosis* pathogenesis. TB pathogenesis involves *M. tuberculosis* infecting alveolar macrophages in the lungs, evading immune clearance, and forming granulomas to persist in a latent state. Progression to active disease can occur when latent infection reactivates, leading to symptomatic tuberculosis and potential transmission to others through respiratory droplets. The image has been taken from the website—https://ntep.in/node/97/KM-pathogenesis-tb (accessed on 8 April 2024) [15].

**Figure 3 vaccines-12-00779-f003:**
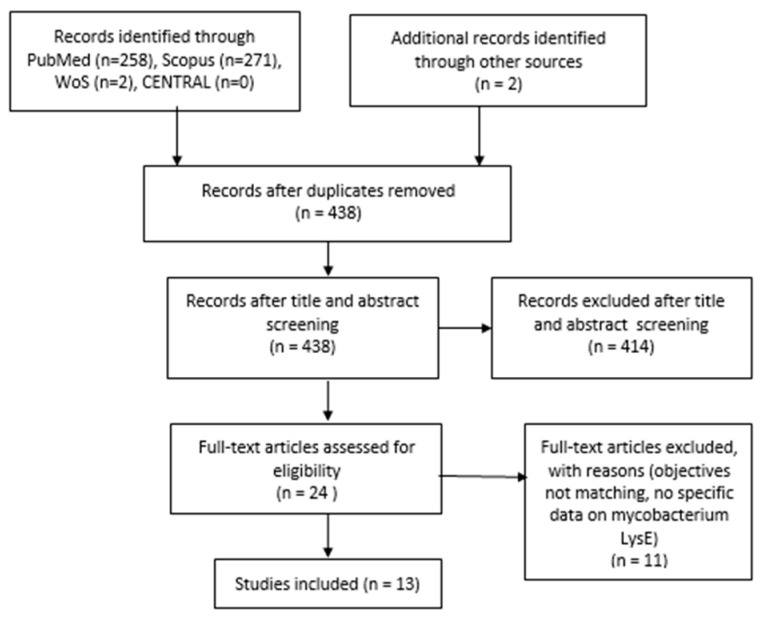
PRISMA Flow Diagram.

**Table 1 vaccines-12-00779-t001:** Research on the multifaceted role of LysE.

Multifaceted Role	Study ID	Findings of the Study	Conclusion
Vaccine Candidate	Georgieva et al. 2020 [21] United States of America (USA)	The study focussed on the *M.tb*-encoded L-lysine exporter (LysE) that serves as a target for inhibiting bacterial growth because its absence results in the accumulation of toxic levels of L-lysine, suppressing bacterial growth. The most studied H37Rv strain of *M.tb* encodes two LysEs, but one of them, the Rv1986 protein, is considered the primary exporter. During the early stages of *M.tb* infection, Rv1986 expression is increased under hypoxia conditions. Thus, the LysE protein can be a probable vaccine candidate. This study developed custom protocols for cloning, expression, purification, and initial characterization of LysE. Protein is purified in lipodiscs made of native *E. coli* membranes and detergent.	This study represented the initial production of highly pure LysE, providing a platform for in-depth investigations into the functional mechanisms of this protein. There is a need to have detailed knowledge about the less studied *M.tb* membrane transport system.
	Chen F. et al. 2018 [8] China	A comprehensive analysis was performed by Chen F. et al. across 12 *M. Bovis* BCG strains and 200 *M. tuberculosis* strains. They proposed the presence of proteins, Rv1986, an amino acid transporter, and an integral membrane sulfoglycolipid transporter Rv3823c, responsible for mycolic acid transport in most epidemic strains but absent in BCG. These proteins exhibited a higher number of T-cell and B-cell epitopes. Notably, Rv1986 is found in virulent *M. Bovis* but not in the BCG strain. The researchers expressed recombinant *M.tb* proteins Rv1986 and Rv3823c in *E. coli*, testing their ability to stimulate innate and adaptive immune cell propagation and cytokine release. The results indicated that Rv1986 and Rv3823c can potentially enhance protective immunity through the activation of T cells, plasma cells, and inflammatory cytokines.	The study suggested the possible roles of Rv1986 and Rv3823c in future vaccine development. Additionally, Rv1986 was identified as an immunodominant target for memory T cells, capable of eliciting both humoral and cell-mediated immune responses, highlighting its diagnostic potential.
	Jiang et al., 2017 [25] China	Region of Deletion 2 (RD2) plays a role in mycobacterial virulence, and its deletion from *M.tb* leads to a decrease in bacterial growth. The researchers illustrated that genes within the RD2 antigens exhibit higher variability than other mycobacterial regions, particularly in the T-cell epitopic region. This implies their potential involvement in genetic variation to evade host immunity. The study specifically involved amplifying and comparing the sequences of five genes within the RD2 region, which encompasses both T and B cell epitopic regions. The results indicate that proteins Rv1980c, Rv1985, and Rv1986 in the RD2 region undergo antigenic variation in response to host immune pressure, suggesting their potential role in ongoing immune evasion.	The data confirmed that RD2 regions, aiming to evade host immunity, exhibit unexpected variability, particularly in immune-related antigens and T-cell epitope regions.
	Gideon et al. 2010 [23] South Africa	Gideon et al. identified the upregulation of two regions of difference (RD) 11 (Rv2658C and Rv2659c) and one RD2 (Rv1986), absent in widely used BCG strains. Exploring gene response patterns during prolonged hypoxia could unveil novel antigens for potential use in vaccines or diagnostics for tuberculosis. The Rv1986 gene exhibited significant elevation among tuberculosis patients during low oxygen levels. In TB patients, analysis of human immune responses to this protein revealed its strong recognition by cells producing interleukin-2 (IL-2), crucial for long-term immunological memory. These findings established Rv1986 as an immunodominant target for memory T cells.	Further examination of BCG strains with or without Rv1986 is essential, holding promise for potentially enhancing the effectiveness of existing or new vaccines or diagnostics for tuberculosis.
	Cockle et al. 2002 [22] United Kingdom	Cockle et al. directed their efforts towards pinpointing more precise reagents that could effectively differentiate between vaccination and infection. Their goal also involved the identification of subunit vaccine candidates to enhance the efficacy of tuberculosis vaccines. This study involved applying comparative genomics and identifying *M. Bovis* and *M. tuberculosis* antigens detection in the BCG vaccine. This study carefully selected 13 open reading frames (ORFs) from the RD1, RD2, and RD14 regions of *M. tuberculosis*. It was investigated in cattle infected with *M. Bovis* and in BCG-vaccinated and unvaccinated control cattle. This examination revealed eight particularly immunogenic antigens (Rv1983, Rv1986, Rv3872, Rv3873, Rv3878, Rv3879c, Rv1979c, and Rv1769).	The potential of these antigens as diagnostic markers or subunit vaccines merits further exploration and study.
Drug target	Dubey et al. 2021 [32] India	This study focused on LysO, which mediates the export of L-lysine and L-thialysine in *E. coli*. They studied the hidden aspect of LysO topology and the mechanism it uses to facilitate export. In *M.tb*, L-Lysine export mediates by LysE, a transmembrane protein. Amino acid exporters are required to arbitrate resistance to toxic analogues of amino acids, which promote translational errors resulting in aberrant protein synthesis. Lack of which can cause bacterial growth retardation	This study presented that amino acid exporters have been characterized primarily at the physiological level. However, there is a notable lack of structural information on this class of membrane proteins, which is crucial for gaining mechanistic insights into the process of amino acid export.
	Georgieva et al. 2020 [27] United States of America (USA)	Georgieva et al. (2020) involved LysE protein purification in Lipodisc, which comprises native *E. coli* membranes and detergents and characterized the LysE gene biophysically. LysE deficiency causes increased cellular levels of L-lysine, which inhibits bacterial growth. Consequently, the LysE proteins found in bacterial pathogens emerge as promising drug targets for inhibition. This underscores the importance of understanding these proteins comprehensively, encompassing their structures and the relationships between structure and function.	This study mentioned that it was the first study on highly pure LysE conducted in a controlled environment. It sets the groundwork for more in-depth investigations into how these proteins function.
	Hasenoehrl et al., 2019 [28] United States of America (USA)	In *M.tb*, lysine permease LysE, encoded by Rv1986, catalyses lysine export. A deficiency of LysE contributes to bacteria’s poor growth. This exporter also has a function in non-auxotrophic settings, as indicated by the M.tb lysE mutant’s delayed escape from the lag phase and sluggish development. LysE is a primary target for memory T cells secreting IL-2 and is believed to contribute to human protective immunity. Resting macrophage and lysosomal exposure models revealed that LysE is involved in human latent TB infection. Amino acids, particularly arginine, have been shown to influence human immunometabolism and T-cell responses. These findings highlight the significance of LysE-mediated export.	These results demonstrate that the aspartate pathway depends on a combination of metabolic control mechanisms in *M.tb*. This pathway is essential for persistence and a promising target for developing anti-tuberculosis drugs. The absence of LysE in eukaryotic cells makes it a probable target for drug discovery.
	Tsu and Saier, 2015 [26] China	In 2015, Tsu and Saier [22] explored the LysE superfamily of transmembrane transport proteins, which facilitate the export of amino acids, lipids, and heavy metal ions. This newly identified superfamily comprises three families: L-lysine and L-arginine exporters (LysE), homoserine/threonine resistance proteins (RhtB), and cadmium ion resistance proteins (CadD). While LysE and RhtB proteins are involved in amino acid ex-port, CadD proteins play a more distant role in cadmium (Cd2+) efflux. Members of the LysE Superfamily contribute to ionic homeostasis, protection against cytoplasmic heavy metals/metabolites, cell envelope assembly, and transmembrane electron flow.	As a result, they impact the physiology and pathogenesis of various microbes, serving as potential targets for drug action. However, many family members remain poorly understood regarding their functional and physiological roles.
	Vrljic et al. 1999 [29] United States of America (USA)	In *C. glutamicum*, Vrljic et al. explained that the LysE carrier protein exhibits remarkable functionality in exporting L-lysine. The investigation delves into the membrane topology of LysE, a protein composed of 236 amino acyl residues, by analysing PhoA- and LacZ-fusions. This study explains that LysE is a family of transmembrane proteins found in other bacterial species, such as *E. coli*, *B. subtilis*, *M. tuberculosis* and *H. pylori*. L-lysine export is required in environments with low concentrations of lysine-containing peptides.	Findings concluded that when the export carrier gene lysE is deleted, exceptionally high cytoplasmic concentrations of L-lysine (>1 M) accumulate, leading to bacteriostasis. Hence, LysE can be a probable drug target.
Diagnostic Marker	Chen J. et al., 2018 [24] China	A study on Region of Deletion 2 (RD2) of M.tb stated that RD2 is responsible for bacterial virulence and showed immunodominancy in infected cattle. Antigens that showed immunogenicity in cattle are Rv1983, Rv1986, Rv1987, and Rv1989c. Out of 87 screened peptides from RD2 proteins, only 10 were recognized by over 10% of active TB patients using a whole blood IFN-γ release assay. The active TB group exhibited significantly elevated IFN-γ release responses to Rv1986-P9, P15, P16, Rv1988-P4, P11, and Rv1987-P11 compared to the control groups (*p* < 0.05). These findings indicate that the six epitopes originating from the RD2 region of *M. tuberculosis* hold promise as potential diagnostic markers for tuberculosis.	This study concluded that six epitopes from RD2, including Rv1986, have potential diagnostic value in active TB. In infected cattle, Rv1983, Rv1986, Rv1987, and Rv1989c within RD2 were identified as immunodominant. This study assessed their diagnostic potential in humans.
	Schneefeld et al., 2017 [30] Germany	Schneefeld et al., 2017 [27] demonstrated that the Rv1985c protein binds to its own and the promoter region of Rv1986. In *M. tuberculosis*, Rv1985c and Rv1986 share identical genomic organization with lysG and lysE in *C. glutamicum*, including a promoter region where both genes overlap. The generation of a precisely defined Rv1985c deletion mutant in M. tuberculosis, coupled with gene expression analysis comparing the wild-type strain and the mutant, demonstrated that the Rv1985c protein activates the transcription of Rv1986 in a lysine-dependent manner. Consequently, it serves as an autoregulatory mechanism for its own expression. Furthermore, whole transcriptome expression analysis revealed that, in addition to lysE(Mt), the Rv1985c protein regulated three other genes, ppsB, ppsC, and ppsD, all involved in cell wall metabolism.	This study delineated the regulatory network of Rv1985c in *M.tb*. Given the resemblance to an orthologous gene pair in *C. glutamicum*, the study proposed renaming Rv1985c to lysG(Mt) and Rv1986 to lysE(Mt). While the function of the Rv1986 protein remains elusive, studies have demonstrated its regulation and recognition by memory T cells in human TB.
	Torres et al. 2015 [31] Mexico	Torres et al. performed a randomized controlled trial including patients with latent tuberculosis infection (LTBI). Patients were randomized into two arms, immediate or delayed isoniazid treatment arms. The research pinpointed the primary inducers of IFN-γ production in individuals with latent tuberculosis infection (LTBI) before isoniazid therapy, identifying Rv0849, Rv1737, and the RD2-encoded Rv1986 (absent in many commonly used BCG strains). Earlier investigations have demonstrated that peptides from Rv1986 exhibit potent stimulation of peripheral cells in both LTBI and active TB patients, inducing notably higher levels of IL2 compared to IFN-γ.	The in vitro IFN-γ responses to these proteins could serve as valuable diagnostic markers for assessing changes linked to the treatment of latent TB infection.

## Data Availability

The original contributions presented in the study are included in the article, further inquiries can be directed to the corresponding author.

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
