# Peer review of "Unveiling the Significance of LysE in Survival and Virulence of Mycobacterium tuberculosis: A Review Reveals It as a Potential Drug Target, Diagnostic Marker, and a Vaccine Candidate"

_vaccines, 2024, doi:10.3390/vaccines12070779_

Round 1

Reviewer 1 Report (Previous Reviewer 2)

Comments and Suggestions for Authors

Dear authors,

the manuscript „Unveiling the Significance of LysE in Survival and Virulence of Mycobacterium tuberculosis: A Review Reveals it as a Potential Drug Target, Diagnostic Marker, and a Vaccine Candidate” after the corrections gives better insight into the potential drug target, diagnostic marker and vaccine candidate. The added table contains a comprehensive information about reviewed articles. It will be adequate for publishing in Vaccines if authors follow the minor reviewers’ comments.

Minor comments:

Line 44: Please add shortcut to “Isoniazid-resistant TB”

Line 64: Write the name of the bacteria with italic.

Line 83: Change “collection” into “group”.

Lines 167-169: Remove crossed sentence.

Author Response

Thank you so much for your valuable comments. Your suggestions have improved the quality of our work. We have incorporated all the minor comments in the latest version of the manuscript in track changes. This peer review process has proved to be a big learning experience for me. Thank you once again for your valuable insights. I hope to get your guidance in future endeavors as well.

Reviewer 2 Report (Previous Reviewer 4)

Comments and Suggestions for Authors

I wish to congratulate the authors for greatly improving the quality of the submission.

Author Response

I would like to add my sincere thanks to the reviewers. Their comments have improved the quality of my work, and I have learned so many things during the peer review process. Thank you once again. I hope to get your guidance in my future endeavors.

This manuscript is a resubmission of an earlier submission. The following is a list of the peer review reports and author responses from that submission.

Round 1

Reviewer 1 Report

Comments and Suggestions for Authors

Summary: The manuscript not only reviews the importance of L-lysine exporter (LysE) in the pathogenesis of Mycobacterium tuberculosis but also underscores the pressing need for further research and development in this area, given its potential as a drug candidate, diagnostic marker, and vaccine candidate.

Comments:

1.       It will be beneficial to add a paragraph about the author's vision on the research and development of LysE's role in Mycobacterium tuberculosis.

2.       I found lots of repetitive text in the manuscript. For example, the text in lines 132-133 is the same as in lines 134-136 and 238-239.

3.       The authors need to amend the abbreviations throughout the manuscript.  For example, Mycobacterium tuberculosis (Mtb) is abbreviated in line 84; however, it was mentioned in the text in line 55. Please review all the abbreviations.

Comments on the Quality of English Language

The Manuscript is well-written in English language. I didn't find obvious grammatical or spelling errors. 

Author Response

Reviewer 1:

Summary: The manuscript not only reviews the importance of L-lysine exporter (LysE) in the pathogenesis of Mycobacterium tuberculosis but also underscores the pressing need for further research and development in this area, given its potential as a drug candidate, diagnostic marker, and vaccine candidate.

Response: Thank you very much for taking the time to review this manuscript and appreciating our work. Please find the detailed responses below and the corresponding revisions highlighted in the re-submitted files

Point-by-point response to Comments and Suggestions for Authors

Comment 1: It will be beneficial to add a paragraph about the author's vision on the research and development of LysE's role in Mycobacterium tuberculosis.

Response 1: Earlier, we added one line about the aim of the study in the last paragraph of the introduction. Now, we have elaborated the vision of the study in a separate paragraph. We have also revised some points in the introduction. Thank you.

Comment 2: I found lots of repetitive text in the manuscript. For example, the text in lines 132-133 is the same as in lines 134-136 and 238-239

Response 2: Thank you for highlighting this. We have removed the repetitive text in the manuscript.

Comment 3: The authors need to amend the abbreviations throughout the manuscript.  For example, Mycobacterium tuberculosis (Mtb) is abbreviated in line 84; however, it was mentioned in the text in line 55. Please review all the abbreviations.

Response 3: We have corrected and rechecked the abbreviations throughout the manuscript. Thank you.

Reviewer 2 Report

Comments and Suggestions for Authors

Dear authors,

the manuscript „Unveiling the Significance of LysE in Survival and Virulence of Mycobacterium tuberculosis: A Review Reveals it as a Potential Drug Target, Diagnostic Marker, and a Vaccine Candidate” is properly written and will be adequate for publishing in Vaccines if authors follow the reviewers’ comments. As authors claims - tuberculosis is a global health threat, necessitating innovative strategies for control and prevention. This review explores the Mycobacterium tuberculosis Ly-17 sine Exporter (LysE) gene, unveiling its multifaceted roles and potential uses in controlling and preventing tuberculosis. As a pivotal player in eliminating excess L-lysine and L-arginine, LysE contributes to the survival and virulence of M. tuberculosis. This review synthesizes findings from different electronic databases. In my opinion Fig 1 and 2 should not be adopted from other authors. Unify the writing of the name and shortcut of bacteria, write the name of bacteria with italic, in lines: 118, 120, 131 (M.tb), 149, 152, 230, 231, 268, 351  and elsewhere.

Minor comments:

Line 85: Please change the word “successful” into “pathogenic”

Line 127: The name of protein should not be written with italic.

Line 148: Correct the name Escherichia coli.

 Line 330: Add information what type of IFN.

Author Response

Reviewer 3:

Summary: Dear authors, the manuscript „Unveiling the Significance of LysE in Survival and Virulence of Mycobacterium tuberculosis: A Review Reveals it as a Potential Drug Target, Diagnostic Marker, and a Vaccine Candidate” is properly written and will be adequate for publishing in Vaccines if authors follow the reviewers’ comments. As authors claims - tuberculosis is a global health threat, necessitating innovative strategies for control and prevention. This review explores the Mycobacterium tuberculosis Lysine Exporter (LysE) gene, unveiling its multifaceted roles and potential uses in controlling and preventing tuberculosis. As a pivotal player in eliminating excess L-lysine and L-arginine, LysE contributes to the survival and virulence of M. tuberculosis. This review synthesizes findings from different electronic databases. In my opinion Fig 1 and 2 should not be adopted from other authors. Unify the writing of the name and shortcut of bacteria, write the name of bacteria with italic, in lines: 118, 120, 131 (M.tb), 149, 152, 230, 231, 268, 351  and elsewhere.

Response: Thank you very much for taking the time to review this manuscript and appreciating our work. Please find the detailed responses below and the corresponding revisions highlighted in the re-submitted files.

According to the copyright act (CC BY 4.0), we can incorporate the image from the open-access article into our work, provided we properly cite the source. However, we have modified the image to meet our requirements. Please find the screenshot.

Thank you for highlighting this. We have now unified and written the bacteria name in Italics throughout the manuscript.

Point-by-point response to Comments and Suggestions for Authors

Comment 1: Line 85: Please change the word “successful” into “pathogenic”.

Response 1: Thank you for pointing this out. Line 114 (85): Changed the word “successful” into “pathogenic”.

Comment 2: Line 127: The name of protein should not be written with italic.

Response 2: Line 157 (127): This is the heading (1.4 LysE, an amino acid transporter protein) as per the journal’s template.

Comment 3: Line 148: Correct the name Escherichia coli.

Response 3: Agree. Line 180 (148): We have now corrected the term.

Comment 4:  Line 330: Add information what type of IFN.

Response 4: We have now incorporated these details in Table 1.

Reviewer 3 Report

Comments and Suggestions for Authors

The paper needs major modifications. The introduction appears as a condensed version of more extensive review monographs. The study design is not appropriate, the materials and methods section is excessive. The descriptions are not thoroughly addressed and only go as far as saying 'potential applications as a drug target, vaccine candidate, and prospective diagnostic marker' without delving deeper into the implications of each concept, the discussion section should be intricately woven into the text, expanding on each of the proposed sections.

Introduction

Line 37. Add the incidence of TB in 2022, based on reference 3. 

Line 42. "In countries with a high burden of TB, such as India". Although some countries have a high burden of MDR-TB, it's important to describe it as heath global problem. Please include the global incidence. 

Line 46-47. Before describing the importance of the BCG vaccine, add a text to entice the importance of research and developing new treatments.

1.1 Overview of Mycobacterium tuberculosis

you should describe that MTBC is composed of several highly genetically related species (>99% nucleotide identity). 

line 65. Based on your description of MTBC and the mention of 'proteins absent in BCG,' readers anticipate that you will cite evidence showing the existence of differences in mycobacterial cell walls that contribute to various aspects of mycobacterial pathogenicity.

1.2 TB Pathogenesis

While the text is engaging, it appears that numerous reviews have already extensively covered this topic. If we are to delve into this subject, I propose incorporating the implications of LysE into our topic. This addition could provide a unique angle and further enrich the content.

2. Materials and Methods

excluding articles containing information about the Lysine-Exporter gene from other bacterial species like C. glutamicum, E. coli, H. pylori, etc., may hinder the integration of knowledge and the comparative exercise to determine knowledge gaps or integration of the same.

3.1 Details of included studies

The descriptions of the selected articles offer nothing more than a description without any discussion or context that may interest the reader.

5. Conclusions

the discussion section should be intricately woven into the text, expanding on each of the proposed sections.

The conclusion is not conclusive; it is necessary to strengthen in the main text the idea of ​​using this protein as a strategy in the TB end plan

Author Response

Reviewer 4:

Summary: The paper needs major modifications. The introduction appears as a condensed version of more extensive review monographs. The study design is not appropriate; the materials and methods section is excessive. The descriptions are not thoroughly addressed and only go as far as saying 'potential applications as a drug target, vaccine candidate, and prospective diagnostic marker' without delving deeper into the implications of each concept, the discussion section should be intricately woven into the text, expanding on each of the proposed sections.

Response: Thank you very much for taking the time to review this manuscript. We have tried to incorporate all your suggestions in the manuscript.

The previous introduction was modified as per the academic editor’s comments to mention about TB pathogenesis and virulence, hence included. The format of the methodology section was adopted from the previous review articles published in the ‘Vaccines’ journal. However, we have now updated the introduction per the reviewer’s comments.

Search strategies were included as a part of the methodology to maintain transparency. Please find the detailed responses below and the corresponding revisions highlighted in the re-submitted files. Thank you.

Point-by-point response to Comments and Suggestions for Authors

Introduction:

Comment 1: Line 37. Add the incidence of TB in 2022, based on reference 3. 

Response 1: Thank you so much for pointing this out. Line 37: We have added the incidence of TB.

Comment 2: Line 42. "In countries with a high burden of TB, such as India". Although some countries have a high burden of MDR-TB, it's important to describe it as heath global problem. Please include the global incidence. Line 46-47. Before describing the importance of the BCG vaccine, add a text to entice the importance of research and developing new treatments.

Response 2: Thank you for your comments. We have included the global incidence in Line 49. We have modified the whole paragraph as per the requirement and included a few lines about the requirement for new treatments. We have also added the information about the research and developing new treatments.

Comment 3: 1.1 Overview of Mycobacterium tuberculosis. you should describe that MTBC is composed of several highly genetically related species (>99% nucleotide identity). line 65. Based on your description of MTBC and the mention of 'proteins absent in BCG,' readers anticipate that you will cite evidence showing the existence of differences in mycobacterial cell walls that contribute to various aspects of mycobacterial pathogenicity.

Response 3: Thank you for your comments. We have incorporated the information as suggested.

Comment 4: 1.2 TB Pathogenesis. While the text is engaging, it appears that numerous reviews have already extensively covered this topic. If we are to delve into this subject, I propose incorporating the implications of LysE into our topic. This addition could provide a unique angle and further enrich the content.

Response 4: Thank you so much for appreciating our work. We have tried to incorporate all the available information about LysE in one review. Unfortunately, very few studies and information were available on LysE to be included. To address this, we have presented search strategies to inform the readers that we have done extensive research and ultimately found the information presented in this review. We have also mentioned this point in the ‘limitation of the study’ section and emphasize the need for more research on LysE.

Materials and methods

Comment 5: excluding articles containing information about the Lysine-Exporter gene from other bacterial species like C. glutamicum, E. coli, H. pylori, etc., may hinder the integration of knowledge and the comparative exercise to determine knowledge gaps or integration of the same.

Response 5: By using the mentioned search strategies in different electronic databases, we got 531 articles. After duplicates were removed, we were left with 438 articles. We screened articles based on the title and abstract before examining full-text articles.

Subsequently, out of the 438 articles initially identified from electronic databases, we excluded 414 articles due to the absence of information concerning lysine exporters or amino acid transporters in the title and abstract of the study, leaving us with 24 articles for further review. Upon thoroughly examining the full texts of these 24 articles, we found that only 13 contained relevant information about the lysine exporter of Mycobacterium tuberculosis.

 While running the search strategies, we may get many articles that include one or two keywords but are not necessarily the articles  that includes the information about LysE of M.tb. As mentioned in the review, this study focuses solely on the Lysine exporter gene of M.tb. Therefore, we excluded the studies that did not focus on the LysE of M.tb.

By this review, we tried to highlight the importance of LysE in M.Tb., which can be a promising vaccine candidate for TB.

3.1 Details of included studies

Comment 6: The descriptions of the selected articles offer nothing more than a description without any discussion or context that may interest the reader.

Response 6: Thank you for pointing this out. I completely agree to the reviewer. We have now incorporated the details of each study in a tabular form that highlights the multifaceted role of LysE along with the study's conclusion. Please find the revised Table 1.

Comment 7: The discussion section should be intricately woven into the text, expanding on each of the proposed sections. The conclusion is not conclusive; it is necessary to strengthen in the main text the idea of ​​using this protein as a strategy in the TB end plan.

Response 7: Thank you for your suggestions. We have modified the discussion and conclusion.

Reviewer 4 Report

Comments and Suggestions for Authors

The authors present a systematic review of the literature on LysE of Mycobacterium Tuberculosis. The objective was to assess its Potential as a drug Target, a diagnostic Marker, and a Vaccine Candidate. The review focused on published reports for the past 25 years.

The identification of new potential drug targets or targets for diagnostic or vaccine development against TB is very important as this disease continues to be of global concern.

While the design of the review is typical of similar reviews, I find two key items as could have been designed better. The first is the selection criteria and the exclusion criteria. Why did the authors chose to not restrict the selection of reports to those covering Mycobacterium tuberculosis from the start, instead of selecting 438 and then exclude 414?

The second point is the results section covering the details of the studies. This section is broken down into individual paragraphs, each presenting a brief summary of each paper. It seems poorly designed and is fragmented. The authors could have done a better job by expanding the table they have in this report to include all 13 studies, with more details of each study. This can then be followed by a detailed analysis of each study and evaluation of the value of the data and conclusions of each study. 

Author Response

Summary: The authors present a systematic review of the literature on LysE of Mycobacterium Tuberculosis. The objective was to assess its Potential as a drug Target, a diagnostic Marker, and a Vaccine Candidate. The review focused on published reports for the past 25 years. The identification of new potential drug targets or targets for diagnostic or vaccine development against TB is very important as this disease continues to be of global concern.

Response: Thank you very much for taking the time to review this manuscript and appreciating our work. We have tried our best to incorporate all your suggestions. Please find the detailed responses below and the corresponding revisions highlighted in the re-submitted files

Point-by-point response to Comments and Suggestions for Authors

Comment 1: While the design of the review is typical of similar reviews, I find two key items as could have been designed better. The first is the selection criteria and the exclusion criteria. Why did the authors chose to not restrict the selection of reports to those covering Mycobacterium tuberculosis from the start, instead of selecting 438 and then exclude 414?

Response 1: Thank you for your comments. We have missed the word ‘title and abstract screening’ in the PRISMA Flow Diagram, although it is referenced in the text. We have now incorporated the modified PRISMA Flow Diagram. My apologies for the same.

As the methodology is concerned, this is how we search the articles while performing ‘Systematic Reviews’ and it is the standard methodology adopted from the ‘Cochrane handbook of Systematic reviews’. By this method, we retrieve the maximum number articles that includes the information of our interest and the chances of missing any relevant article is negligible. As we search the articles by using relevant keywords present in the title and abstract, hence the initial number of articles would be high.

By using the mentioned search strategies in different electronic databases, we got 531 articles. After removing the duplicates, we were left with 438 articles. We screened articles based on the title and abstract before examining the full-text articles. This is called the initial screening of the articles.

Subsequently, out of the 438 articles initially identified from electronic databases, we excluded 414 articles due to the absence of information concerning lysine exporters or amino acid transporters in the title and abstract of the study, leaving us with 24 articles for further review. Upon thoroughly examining the full texts of these 24 articles, we found that only 13 contained relevant information about the lysine exporter of Mycobacterium tuberculosis.

Comment 2: The second point is the results section covering the details of the studies. This section is broken down into individual paragraphs, each presenting a brief summary of each paper. It seems poorly designed and is fragmented. The authors could have done a better job by expanding the table they have in this report to include all 13 studies, with more details of each study. This can then be followed by a detailed analysis of each study and evaluation of the value of the data and conclusions of each study. 

Response 2: Thank you so much for your suggestion. I completely agree with the reviewer, We have expanded Table 1 by incorporating all the 13 included studies and removed the individual paragraphs.